# Effect of Black Pepper (*Piper nigrum*) Extract on Caffeine-Induced Sleep Disruption and Excitation in Mice

**DOI:** 10.3390/nu14112249

**Published:** 2022-05-27

**Authors:** Minseok Yoon, Jonghoon Jung, Minjung Kim, Changho Lee, Suengmok Cho, Minyoung Um

**Affiliations:** 1Research Division of Food Functionality, Korea Food Research Institute, Wanju 55365, Korea; msyoon@kfri.re.kr (M.Y.); jonghoon@kfri.re.kr (J.J.); mjkim14@kfri.re.kr (M.K.); chang@kfri.re.kr (C.L.); 2Department of Food Science and Technology, Institute of Food Science, Pukyong National University, Busan 48513, Korea; scho@pknu.ac.kr; 3Division of Food Biotechnology, University of Science & Technology, Daejeon 34113, Korea

**Keywords:** black pepper, *Piper nigrum*, caffeine-induced sleepiness, sleep-promoting, hyperlocomotion

## Abstract

Sleep is one of the most essential factors required to maintain good health. However, the global prevalence of insomnia is increasing, and caffeine intake is a major trigger. The objective of this study was to investigate the inhibitory effect of black pepper, *Piper nigrum* extract (PE), on caffeine-induced sleep disruption and excitation in mice. Caffeine significantly decreased sleep duration in the pentobarbital-induced sleep test. It also resulted in a significant increase in sleep onset and a decrease in non-rapid eye movement sleep. Moreover, in an open-field test, caffeine-treated mice exhibited a significantly increased time in the center zone and total distance traveled. However, the co-administration of caffeine and PE did not result in similar arousal activities. Thus, our results suggest that PE can be used as a potential therapeutic agent to treat sleep problems and excitatory status associated with caffeine intake.

## 1. Introduction

Sleep is one of the most essential physiological phenomena for maintaining health [1,2]. It is easy to understand the importance of sleep just by noting that we sleep for one-third of our lifespan [3,4,5]. Insomnia is typically characterized by unrecoverable sleep and daytime dysfunction due to difficulty in initiating or maintaining sleep [6]. It adversely affects physical and mental health, including the cardiovascular and immune systems, memory, learning, and mood [7,8]. Therefore, insomnia is a major cause of health and quality of life deterioration. It also causes accidents, such as drowsy driving and reduced workability, which incur significant social costs. Despite the importance of sleep, insomnia has recently become more widespread in developed countries. Globally, approximately 10–15% of adults suffer from chronic insomnia, and approximately 25–35% suffer from intermittent insomnia [9]. It has emerged as a major public health concern [10]. Recently, the prevalence of sleep problems has increased owing to the coronavirus disease 2019 pandemic [7,11]. As insomnia becomes more common, herbal sleep aids are gaining global popularity as an alternative to prescription drugs for the treatment of insomnia [12]. Thus, many studies have been conducted on the sleep-improvement effects of natural products.

Caffeine is one of the most consumed food ingredients worldwide and is mainly used in caffeinated beverages, including coffee and energy drinks [13]. It is widely used as a stimulant to enhance arousal [14]. Caffeine intake is a well-known cause of insomnia in humans and rodents. For example, caffeine reduces sleep efficiency and increases sleep onset in normal young adults (18–30 years old) [15]. Furthermore, it markedly increases wakefulness and decreases non-rapid eye movement sleep (NREMS) in animal models [16,17,18]. Caffeine also acts on the central nervous system to induce excitation [19]. These effects of caffeine may have a beneficial impact on daytime sleepiness; however, they cause sleep disruption the following night [20]. Despite the increase in caffeine consumption worldwide, there are very few studies on food products that attenuate caffeine-induced sleep disturbances. Foods and food constituents that can inhibit the arousal effects of caffeine are considered to have potential as natural sleep aids [16,21].

Black pepper (*Piper nigrum*) is widely used worldwide as a spice and has a massive trade share in the global market [22]. Black pepper exhibits various biological properties, including anticancer [23], analgesic [24], anticonvulsant [25], antidepressant [26], and hepatoprotective [27] effects. Moreover, black pepper exerts a hypnotic effect by increasing hexobarbital sleeping time [28]. Piperine, a major alkaloid isolated from black pepper, increases pentobarbitone sleeping time [29] and has been identified as a positive allosteric modulator of γ-aminobutyric acid type A (GABA_A_) benzodiazepine receptor in in vitro cell models [30,31]. Despite these documented benefits, detailed studies evaluating the effect of *Piper nigrum* extract (PE) on caffeine-induced sleep disruption and motor behavior have not yet been conducted. In the previous study, it was reported that phlorotannin, a brown algae polyphenol acting on GABA_A_ benzodiazepine receptor, inhibited the caffeine-induced arousal effect in mice [16]. Furthermore, Ko et al. [21] reported that evodiamine, a major component of *Evodia rutaecarpa*, reduces sleep disruption and hyperlocomotion caused by caffeine through GABA_A_–ergic system. Based on these facts, we hypothesized that PE containing piperine, which acts on the GABA_A_ benzodiazepine receptor, could inhibit caffeine-induced arousal and hyperlocomotion. The present study aimed to evaluate the effects of PE on the sleep–wake profile through analysis of sleep architecture and locomotor activity in a caffeine-induced sleep disruption mouse model.

## 2. Materials and Methods

### 2.1. Reagents and Chemicals

Piperine and caffeine were purchased from Sigma–Aldrich (St. Louis, MO, USA). Zolpidem (ZPD), used as a reference hypnotic, was obtained from the Ministry of Food and Drug Safety (Cheongju, Korea). Pentobarbital was purchased from Hanlim Pharm. Co., Ltd. (Seoul, Korea). All the other reagents and chemicals were of the highest commercially available grade.

### 2.2. Preparation of PE and Liquid Chromatography with Tandem Mass Spectrometry Analysis

For extract preparation, *Piper nigrum* was extracted in a 10 × 70% ethanol aqueous solution at 50 °C for 16 h. The extracts were then filtered and concentrated under reduced pressure and then lyophilized. Piperine is a major active compound in *Piper nigrum*, and liquid chromatography with tandem mass spectrometry (LC–MS/MS) was used to quantify it in PE using an Agilent Technologies system (including a binary pump, wellplate sampler, and thermostatic column compartment) with an Agilent 6490 triple quadrupole mass spectrometer (Agilent Inc., Santa Clara, CA, USA). Piperine was resolved on a Phenomenex Kinetex Phenyl-hexyl column (50 mm × 2.1 mm × 2.6 μm, Torrance, CA, USA). The mobile phase gradient started with 0.1% formic acid in distilled water: 0.1% formic acid in acetonitrile (95:5) for 30 s, followed by a linear increase to 100% of 0.1% formic acid in acetonitrile for 10 min (held for 1 min and then equilibrated for 9 min using the initial condition). Piperine was detected using MS/MS operated in positive ion electrospray ionization and multiple reaction monitoring mode using the transition of m/z 286.1→201.0. The MS parameters used for piperine analysis were as follows: gas temperature, 300 °C; gas flow, 11 L/min; nebulizer, 15 psi; and capillary voltage, 4000 V. The presence of piperine was confirmed by comparing the peaks in the samples with the respective peaks for standard piperine at a comparable retention time. The retention time of piperine was 6.6 min (Figure 1). The concentration of piperine in PE was 24.0 ± 0.9 mg/g extract.

### 2.3. Animals and Treatments

All procedures on animals were performed in agreement with the animal care and use guidelines of the Korea Food Research Institutional Animal Care and Use Committee (permission numbers: KFRI-M-20019 and KFRI-M-20021) and carried out following the ARRIVE guidelines 2.0. C57BL/6N mice (male, 11 weeks old) and Institute of Cancer Research (ICR) mice (male, 3 weeks old) were purchased from Koatech Animal, Inc. (Pyeongtaek, Korea). All animals were housed in an insulated, sound-proof recording room maintained at an ambient temperature of 23 ± 0.5 °C, with a constant relative humidity (55 ± 2%) on an automatically controlled 12 h light/dark cycle (lights off at 17:00). The animals were acclimatized for one week before use. Immediately before oral administration to mice, all samples were dissolved in saline containing 5% Tween 80. Caffeine and ZPD were also administered in the same manner.

### 2.4. Pentobarbital-Induced Sleep Accelerated Test

The pentobarbital-induced sleep test was performed according to the method by Yang et al. [32]. Briefly, ICR mice were used after fasting for 24 h prior to the experiment, and all experiments were performed between 13:00 and 17:00. Mice were orally administered samples 45 min before the test and pentobarbital (45 mg/kg) was administered intraperitoneally (i.p.). The sleep duration of pentobarbital-treated mice was measured and defined as the time difference between loss and recovery of righting reflexes.

### 2.5. Sleep–Wake Profile Analysis

Sleep architecture was analyzed using the method described by Um et al. [33]. A head mount (Pinnacle Technology, Inc., Lawrence, KS, USA) was chronically implanted in mice under pentobarbital anesthesia (50 mg/kg) to record electroencephalogram (EEG) and electromyogram (EMG) signals. Before surgery, the heads and necks of anesthetized mice were shaved and cleaned with ethanol for disinfection. The head of an anesthetized mouse was aligned in a stereotaxic frame (David Kopf Instruments, Los Angeles, CA, USA) and then the skin was incised to expose the skull surface. After placing the front edge of the head mount 3.0 mm anterior to the bregma of the dry skull, four electrode screws for EEG recording were inserted. Two EMG wires were placed bilaterally into a small pocket made in the nuchal muscles using a pair of forceps. Finally, the head mount was securely fixed onto the skull using dental cement. After surgery, the mice were transferred to individual cages for recovery for at least a week. Then, they were adapted for four days under the recording conditions before the experiment. EEG and EMG were recorded using a slip ring designed to allow free behavioral movement of mice. A PAL-8200 data acquisition system (Pinnacle Technology, Inc., Lawrence, KS, USA) was used to record the EEG and EMG signals. The conditions used for recording signals were as follows: sampling rate, 200 Hz; amplifying action, 100×; low-pass filter, 10 Hz for EEG and 10 Hz for EMG. Sleep–wake states were monitored for 48 h, which included baseline data acquisition and experimentation. The baseline was conducted for a period of 24 h in each animal, beginning at 17:00 h. These baseline recordings were used as the control for the corresponding animals. Mice were considered to be falling asleep when there was no detectable signal in EMG. For sleep-stage analysis, EEG and EMG signals were divided into 10 s epochs. SleepSign software version 3.0 (Kissei Comtec, Nagano, Japan) automatically classifies the vigilance states as wakefulness (Wake), REMS, and NREMS. At the end of each predefined stage, signals were visually inspected, and corrected if necessary. Sleep latency was measured by the time taken for the first NREMS episode to appear (lasting for at least 120 s) from the time of drug administration. Bouts of each stage were defined as the periods of one or more consecutive 10 s epochs.

### 2.6. Open Field Test (OFT)

According to Lim et al. [34], we measured hyperlocomotion using the OFT. One hour before each test, samples were orally administered to mice. They were placed in the center of the open field arena (50 × 50 × 50 cm) and their locomotor-related behaviors were recorded for 5 min. We used the video tracking system SMART v3.0 (Panlab SL, Barcelona, Spain) to analyze the times in the center zone and the total distance.

### 2.7. Data Analysis

All data were analyzed using the Prism 5.0 software (GraphPad Software, Inc., San Diego, CA, USA) and presented as the mean ± standard error of the mean (SEM). Data were analyzed using one-way ANOVA, followed by Tukey’s test for multiple comparisons. Comparisons between two-group data were analyzed using the paired Student’s *t*-test. *p*-values of less than 0.05 were considered statistically significant for all tests.

## 3. Results

### 3.1. PE Attenuates Caffeine-Induced Sleep Disruption in ICR Mice

We investigated whether PE could attenuate sleep disturbance induced by caffeine. In this study, caffeine (50 mg/kg) significantly (*p* < 0.05) inhibited pentobarbital-induced sleep in ICR mice (Figure 2). Co-administration of PE at 100, 250, or 500 mg/kg with caffeine significantly (*p* < 0.01) increased sleep duration compared to treatment with caffeine alone. Thus, PE can attenuate the wake-enhancing effects of caffeine.

### 3.2. Effects of PE on Sleep Onset, Total Sleep Time, and Sleep Architecture of C57BL/6N Mice in Caffeine-Induced Sleep Disruption Model

To identify additional hypnotic effects of PE, we analyzed the sleep architecture of C57BL/6N mice based on EEG and EMG recordings. The dosages of caffeine (25 mg/kg), PE (500 mg/kg), and ZPD (10 mg/kg) were chosen based on preliminary experiments. Figure 3 presents the sleep–wake profiles of mice after oral administration of caffeine alone, caffeine + PE, and caffeine + ZPD. The value of sleep onset was 189.5 ± 21.3 min in mice administered caffeine alone (Figure 3a). This value increased significantly (*p* < 0.01) after vehicle treatment (57.7 ± 2.9 min). However, there were no significant differences in sleep onset between oral administration of caffeine + PE and caffeine + ZPD and each vehicle, respectively. We also calculated the total sleep time during the dark periods (Figure 3b). During the dark period, the total sleep time in mice treated with caffeine alone was 211.5 ± 30.0 min, which was significantly (*p* < 0.01) shorter than that in the vehicle group (300.8 ± 23.9 min). On the other hand, caffeine + PE and caffeine + ZPD appeared similar to each vehicle and significantly increased total sleep time compared with caffeine alone. Additionally, the duration of each stage during the first 5 h after the administration of caffeine alone, caffeine + PE, and caffeine + ZPD were calculated (Figure 3c). The caffeine increased the amount of Wake by 1.16-fold and reduced the amount of NREMS by 2.02-fold compared with the vehicle. Caffeine + PE or caffeine + ZPD groups presented significantly (*p* < 0.01) shorter Wake and longer NREMS than the caffeine alone group. No significant difference in REMS was found in any group.

Figure 4 shows the temporal change in NREMS, REMS, and Wake for 24 h after the administration of caffeine alone, caffeine + PE, and caffeine + ZPD. The duration of NREMS from the first to the fifth hour after caffeine administration was significantly decreased by 12.4, 4.4, 1.9, 1.6, and 1.6 times, respectively, compared to each vehicle, and at the eleventh hour it was also reduced by approximately 1.6 times (Figure 4a). Concomitant with this decrease in NREMS, the duration of Wake increased during the same time period. Unlike treatment with caffeine alone, the co-administration of PE or ZPD with caffeine did not change the sleep architecture for 24 h (Figure 4b,c).

### 3.3. Effects of PE and ZPD on the Characteristics of Sleep–Wake Episodes of C57BL/6N Mice in Caffeine-Induced Sleep Disruption

To better characterize the arousal-inhibitory effects of PE and ZPD, we additionally analyzed the mean duration and total number of the sleep phase episodes (Figure 5). Caffeine significantly increased the mean duration of Wake by 306.43 s (*p* < 0.01). Conversely, there was no change in NREMS and REMS (Figure 5a). Caffeine also reduced the number of both Wake and NREMS bouts by 2.65-fold (Figure 5b). In contrast, in the co-administration of PE or ZPD with caffeine, no changes were observed. 

### 3.4. Effects of PE on Hyperlocomotion for Caffeine-Treated ICR Mice

We determined the effect of PE on caffeine-induced excitation in ICR mice using an OFT. Caffeine (50 mg/kg)-treated mice exhibited a significantly (*p* < 0.01) increased time in the center zone and total distance traveled (Figure 6a,b). Treatment with PE (250 and 50 mg/kg) significantly attenuated this behavioral effect on excitation compared with caffeine treatment. Finally, we summarize the experimental results in Table 1.

## 4. Discussion

Caffeine is a major psychoactive constituent [35]. According to Lee et al. [36], the average caffeine consumption in South Korea is approximately 81.91 mg/person/day. Unfortunately, caffeine, even at relatively low doses, causes several adverse behavioral effects, including convulsive activity, arousal, and locomotor stimulation [21]. Additionally, caffeine intake to improve daily performance is usually responsible for sleep disturbances and insomnia [37]. In general, caffeine administration induces brain excitation by inhibiting adenosinergic receptors. Caffeine-induced activation of locomotion, hyperthermia, and sleep disruption has been shown to antagonize the functional effects of adenosine agonists acting on A_1_ and/or A_2A_ receptors [38,39]. Therefore, the widely used caffeine-induced arousal model is suitable for recapitulating the fundamental features of insomnia [40,41]. The sleep-disturbing effects of caffeine have been reported by Kwon et al. [16] and Cho et al. [41]. Consistent with these reports, as expected, caffeine significantly decreased sleep duration in a pentobarbital-induced sleep test in the present study. In addition, caffeine caused a significant increase in sleep onset and a decrease in NREMS during the first 5 h after administration. These results indicated that sleep deprivation was well induced by caffeine.

In the previous study, substances such as phlorotannin, evodiamine, and *Nelumbo nucifera* extract that act on GABA_A_ receptors inhibit the arousal and hyperlocomotion caused by caffeine [16,17,21]. In addition, piperine acts on GABA_A_ receptors that induce sleep [42]. Therefore, in this study, the effects of PE were evaluated in the caffeine-induced sleepiness model and compared with those of ZPD, a positive control. ZPD is also known to act on GABA_A_-benzodiazepine receptors [43]. The co-administration of ZPD and caffeine has been reported to significantly inhibit caffeine-induced increase in sleep onset in animals [44]. Additionally, sedative-hypnotic agents such as ZPD or trazodone attenuate caffeine-induced arousal in humans [45]. In this study, the co-administration of PE or ZPD with caffeine did not change sleep onset or NREMS compared to each vehicle. These results suggest that the sleep-promoting effects of PE and ZPD were attenuated by caffeine-induced arousal in mice. Thus, these results imply that PE counteracts the sleep-disturbing effects of caffeine but does not compete for binding to adenosine receptors of caffeine, similar to ZPD.

To further elucidate the inhibitory effect of PE on caffeine-induced hyperlocomotion, we performed an OFT. The time in the center zone of the open field is an indicator of anxiety, and the total distance traveled provides the degree of general activity [46]. Caffeine is correlated with excitation-related behavior. For instance, caffeine intake at low doses induces hyperlocomotion [47]. Caffeine has general excitation effects via antagonizing adenosine receptors [48]. Moreover, the GABAergic system plays a direct role in regulating motor activity through the interaction of dopamine and adenosine [49]. Our results showed that caffeine-treated mice exhibited significantly increased time in the center zone and total distance traveled in the OFT, indicating greater excitation. The caffeine-induced effects on hyperlocomotion were significantly reduced after PE administration. These results suggest that PE counteracts the excitation caused by caffeine.

Our study has some limitations. First, although neurotransmitters such as adenosine, dopamine, and GABA and its receptors play an important role in sleep regulation, we did not measure them in brain tissues of caffeine treated mice. Therefore, further study is needed to explore the molecular mechanism of PE on caffeine-induced sleep disruption. Second, it is assumed that piperine may have contributed to the effects of PE, but further work, such as metabolite analysis, in the brain is required to clarify.

## 5. Conclusions

In this study, we found that PE attenuated caffeine-induced sleep disruption and hyperlocomotion, similar to the hypnotic ZPD. Considering the global trend of increasing consumption of coffee and energy drinks, PE, with an attenuating effect on sleep disturbance and excitation due to caffeine, may have added value as a sedative-hypnotic natural agent.

## Figures and Tables

**Figure 1 nutrients-14-02249-f001:**
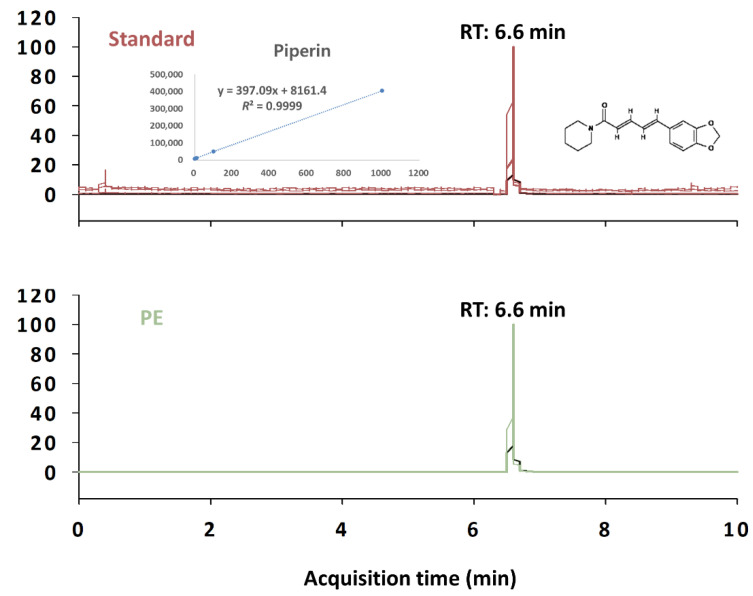
Representative LC–MS/MS of piperine in *Piper nigrum* extract (PE). RT: retention time.

**Figure 2 nutrients-14-02249-f002:**
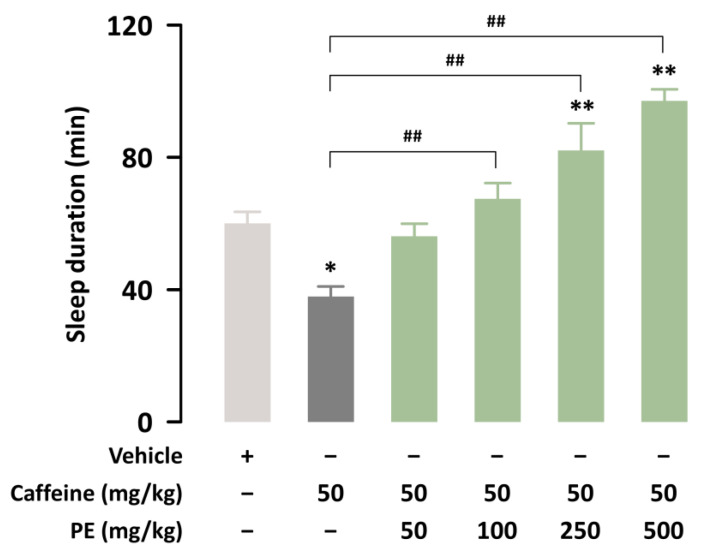
Effects of co-administration of PE and caffeine on sleep duration in pentobarbital-treated ICR mice. Each value represents the group mean ± SEM (*n* = 10 per group). * *p* < 0.05, ** *p* < 0.01, significant difference compared to the vehicle group (Tukey’s test). ## *p* < 0.01, significant difference compared to the caffeine-alone-treated group.

**Figure 3 nutrients-14-02249-f003:**
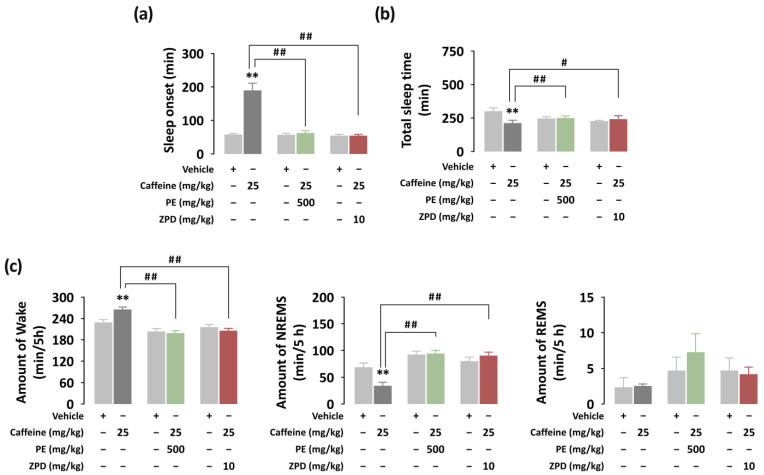
Sleep–wake profiles in C57BL/6N mice after oral administration of caffeine alone, caffeine + PE, and caffeine + ZPD. (**a**) Effects of caffeine alone, caffeine + PE, and caffeine + ZPD on sleep onset. (**b**) Effects of caffeine alone, caffeine + PE, and caffeine + ZPD on total sleep time during the dark period. (**c**) The duration of Wake, NREMS, and REMS during the 5 h period after the administration of vehicle, caffeine, caffeine + PE, and caffeine + ZPD. Each value represents the mean ± SEM (*n* = 7–8 per group). ** *p* < 0.01, significantly different from the vehicle control (paired Student’s *t*-test) and # *p* < 0.05, ## *p* < 0.01, significant difference between two groups (unpaired Student’s *t*-test).

**Figure 4 nutrients-14-02249-f004:**
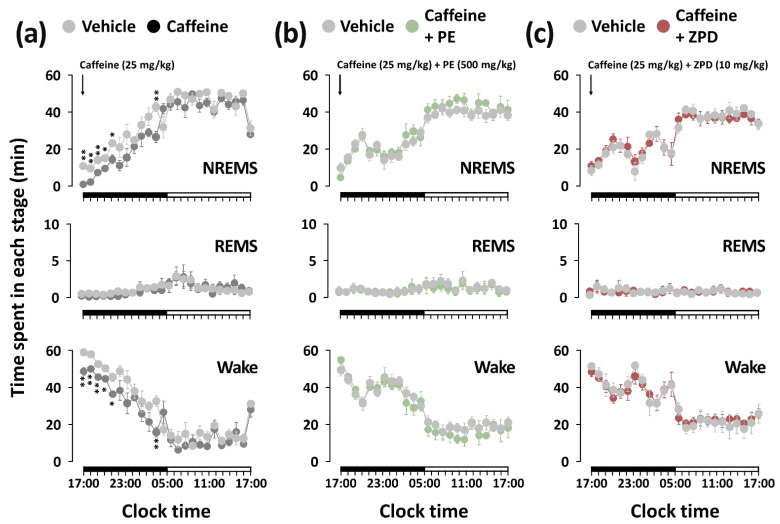
Temporal changes in NREMS, REMS, and Wake for 24 h after oral administration of caffeine alone (**a**), caffeine + PE (**b**), or caffeine + ZPD (**c**) to C57BL/6N mice. Each circle represents the hourly mean ± SEM amount of each stage (*n* = 7–8 per group). * *p* < 0.05 and ** *p* < 0.01, significantly different from the vehicle control (paired Student’s *t*-test).

**Figure 5 nutrients-14-02249-f005:**
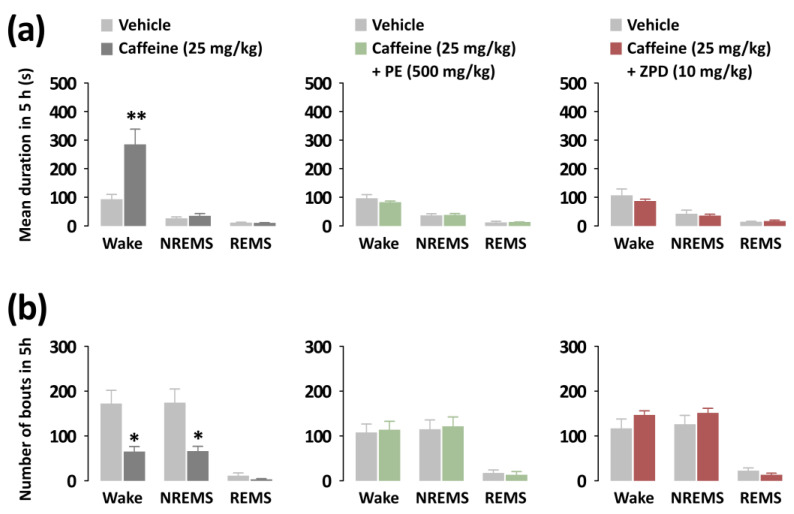
Characteristics of sleep–wake episodes in C57BL/6N mice during the 5 h period after the administration of caffeine alone, caffeine + PE, or caffeine + ZPD. (**a**) Changes in the mean duration of wake, NREMS, and REMS bouts. (**b**) Changes in the total number of Wake, NREMS, and REMS bouts. Each value represents the mean ± SEM (*n* = 7–8 per group). * *p* < 0.05, ** *p* < 0.01, significantly different from the vehicle control (paired Student’s *t*-test).

**Figure 6 nutrients-14-02249-f006:**
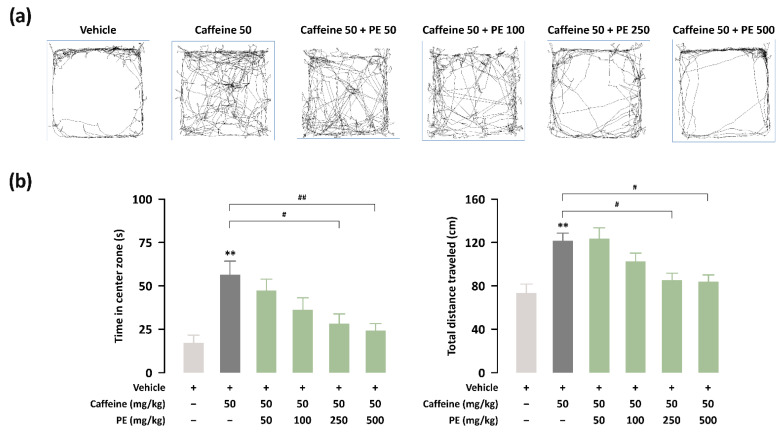
Effect of PE on the open field test (OFT) in caffeine-induced excitation. (**a**) Individual examples of locomotor activity. (**b**) Time in the center zone (s) and total distance traveled (cm) in the OFT (*n* = 10 per group). Results are presented as mean ± SEM. ** *p* < 0.01, significant difference compared to the vehicle group (Tukey’s test). # *p* < 0.05, ## *p* < 0.01, significant difference compared to the caffeine-alone-treated group.

**Table 1 nutrients-14-02249-t001:** Summary of the main findings in the present study.

Methodology	Parameter	PE Treatment ^(1)^
Pentobarbital-induced sleep test	Sleep duration	Increased
Sleep–wake profile analysis	Sleep onset (min)	Decreased
	Total sleep time (min)	Increased
	Wake amount (min/5 h)	Decreased
	NREMS amount (min/5 h)	Increased
	REMS amount (min/5 h)	No change
Open field test	Locomotor activity	Decreased

^(1)^ Compared with caffeine treated group.

## Data Availability

Not applicable.

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
