# Peer review of "Effect of Black Pepper (Piper nigrum) Extract on Caffeine-Induced Sleep Disruption and Excitation in Mice"

_nutrients, 2022, doi:10.3390/nu14112249_

Round 1

Reviewer 1 Report

This is interesting sleep study using animal model. However, I found the following major and minor flaws:

First general comment - when Authors provide a big fact they have to cite reliable papers from reputable journals not only a "small players".

  1. Please cite theses articles: Matricciani L, Paquet C, Galland B, Short M, Olds T. Children's sleep and health: A meta-review. Sleep Med Rev. 2019 Aug;46:136-150. doi: 10.1016/j.smrv.2019.04.011. Epub 2019 Apr 23. PMID: 31121414. Chaput JP, Dutil C, Featherstone R, Ross R, Giangregorio L, Saunders TJ, Janssen I, Poitras VJ, Kho ME, Ross-White A, Zankar S, Carrier J. Sleep timing, sleep consistency, and health in adults: a systematic review. Appl Physiol Nutr Metab. 2020 Oct;45(10 (Suppl. 2)):S232-S247. doi: 10.1139/apnm-2020-0032. PMID: 33054339. after the sentence: Sleep is one of the most essential physiological phenomena for maintaining health.
  2. Please cite these articles: Meredith Wallace, Nicholas Kissel, Martica Hall, Anne Germain, Karen Matthews, Wendy Troxel, Peter Franzen, Daniel Buysse, Kathryn Roecklein, Heather Gunn, Brant Hasler, Tina Goldstein, Dana McMakin, Eva Szigethy, Adriane Soehner, 540 Age Trends in Sleep Across the Lifespan: Findings from the Pittsburgh Lifespan Sleep Databank, Sleep, Volume 44, Issue Supplement_2, May 2021, Page A213, https://doi.org/10.1093/sleep/zsab072.538 Sigga Svala Jonasdottir, Kelton Minor, Sune Lehmann, Gender differences in nighttime sleep patterns and variability across the adult lifespan: a global-scale wearables study, Sleep, Volume 44, Issue 2, February 2021, zsaa169, https://doi.org/10.1093/sleep/zsaa169Marissa A Evans, Daniel J Buysse, Anna L Marsland, Aidan G C Wright, Jill Foust, Lucas W Carroll, Naina Kohli, Rishabh Mehra, Adam Jasper, Swathi Srinivasan, Martica H Hall, Meta-analysis of age and actigraphy-assessed sleep characteristics across the lifespan, Sleep, Volume 44, Issue 9, September 2021, zsab088, https://doi.org/10.1093/sleep/zsab088after the sentence "It is easy to understand the importance of sleep just by noting that we sleep for one third of our lifespan".
  3. Please cite this article Fila-Witecka K, Malecka M, Senczyszyn A, Wieczorek T, Wieckiewicz M, Szczesniak D, Piotrowski P, Rymaszewska J. Sleepless in Solitude-Insomnia Symptoms Severity and Psychopathological Symptoms among University Students during the COVID-19 Pandemic in Poland. Int J Environ Res Public Health. 2022 Feb 23;19(5):2551. doi: 10.3390/ijerph19052551. PMID: 35270244; PMCID: PMC8909406 after the sentences:  "Recently, the prevalence of sleep problems has increased owing to the coronavirus disease
    2019 pandemic." and  "It adversely affects physical and mental health, including the cardiovascular and immune systems, memory, learning, and mood." and this article

    Martynowicz H, Skomro R, Gać P, Mazur G, Porębska I, Bryłka A, Nowak W, Zieliński M, Wojakowska A, Poręba R. The influence of hypertension on daytime sleepiness in obstructive sleep apnea. J Am Soc Hypertens. 2017 May;11(5):295-302. doi: 10.1016/j.jash.2017.03.004. Epub 2017 Mar 27. PMID: 28412276.

    after the sentence "It adversely affects physical and mental health, including the cardiovascular and immune systems, memory, learning, and mood."
  4. Please cite the following article: Lin YN, Liu ZR, Li SQ, Li CX, Zhang L, Li N, Sun XW, Li HP, Zhou JP, Li QY. Burden of Sleep Disturbance During COVID-19 Pandemic: A Systematic Review. Nat Sci Sleep. 2021 Jun 28;13:933-966. doi: 10.2147/NSS.S312037. PMID: 34234598; PMCID: PMC8253893 after the sentence "Recently, the prevalence of sleep problems has increased owing to the coronavirus disease 2019 pandemic."
  5. Authors have to provide a null hypothesis at the end of Introduction.
  6. Authors have to clearly emphasized in the Introduction why their study is timely and innovative and what was the reason to performed this study. Furthermore, Authors have to present results of similar studies related to the topic.
  7. Authors have to provide manufacturer and country of origin for each device, set, substance etc applied in the study within Materials and Methods.
  8. Authors have to provide an information about performing or not performing study in accordance to the ARRIVE guidelines https://www.nc3rs.org.uk/arrive-guidelines
  9. Authors have to provide detailed informations about sleep-wake recording using EEG and EMG. The description provided in 2.5 section is limited. Please remember that this recording is a base to accept or reject the null hypothesis and a core for this study.
  10. Authors have to design a summarizing table which will present the main findings and add it to the Results section.
  11. Authors have to strongly confront their findings with similar studies in the Discussion.

Reviewer 2 Report

This study evaluated the effects of Black pepper (Piper nigrum) Extract on the sleep-wake profile using an analysis of sleep architecture and locomotor activity in a caffeine-induced sleep disruption mouse model. It was confirmed that PE attenuated caffeine-induced sleep disruption and hyperlocomotion. This is an interesting discovery, the following problems need to be solved:

  1. Why were two different breeds of mice of different ages selected for the experiment in this study? And the description of animal experiments methods is not clear, nor the number of mice in each experimental group.
  2. The settings of Vehicle groups in different experiments are inconsistent, making the experimental data difficult to understand.
  3. The sleep duration of ICR mice after 500mg/kg PE intervention was significantly higher than that of the Vehicle group. Why was it different in C57BL/6N mice?
  4. The detection of adenosine agonist, dopamine and other indicators in mice is necessary to analyze the role of PE in alleviating caffeine-induced sleep disruption.

Round 2

Reviewer 1 Report

Good job! The manuscript has been significantly improved and it looks perfect now.

I recommend to accept the manuscript unaltered.

Reviewer 2 Report

The authors answered all my questions and the manuscript has been much improved.